# Can the combination of antiplatelet or alteplase thrombolytic therapy with argatroban benefit patients suffering from acute stroke? a systematic review, meta-analysis, and meta-regression

Haiyan Xie[1], Ying Chen[2], Wukun Ge[3], Xiuping Xu[4]*, Chengjiang Liu[5], Zhiyong Lan[6], Yina Yang[7]

1 Department of Clinical Pharmacy, The Third Hospital of Quzhou, Quzhou, Zhejiang, 324003, China, 2 Hangzhou Fuyang District Hospital of Traditional Chinese Medicine, Zhejiang, 311499, China, 3 Department of Clinical Pharmacy, Ninghai First Hospital, Zhejiang, 315600, China, 4 Department of Emergency Internal Medicine, The Third Hospital of Quzhou, Quzhou, Zhejiang, 324003,China, 5 Department of General Medicine, Affiliated Anqing First People's Hospital of Anhui Medical University, HeFei, 246000, China, 6 Department of Psychiatry Department, The Third Hospital of Quzhou, Quzhou, Zhejiang, 324003, China, 7 Department of Neurology, Ninghai First Hospital, Zhejiang, 315600, China

* 15695701156@163.com

**Data Availability Statement:** All relevant data are within the manuscript and its Supporting Information files.

## Abstract

### Background

The effectiveness of administering argatroban as a treatment approach following antiplatelet therapy or alteplase thrombolytic therapy in patients with acute stroke is presently uncertain. However, it is important to highlight the potential benefits of combining this medication with known thrombolytics or antiplatelet therapy. One notable advantage of argatroban is its short half-life, which helps minimize excessive anticoagulation and risk of bleeding complications in inadvertent cases of hemorrhagic stroke. By conducting a meticulous review and meta-analysis, we aim to further explore the common use of argatroban and examine the plausible advantages of combining this medication with established thrombolytic and antiplatelet therapies.

### Method

In this study, we performed a rigorous and methodical search for both randomized controlled trials and retrospective analyses. Our main objective was to analyze the impact of argatroban on the occurrence of hemorrhagic events and the mRS scores of 0–2. We utilized a meta-analysis to assess the relative risk (RR) associated with using argatroban versus not using it.

### Results

In this study, we analyzed data from 11 different studies, encompassing a total of 8,635 patients. Out of these patients, 3999(46.3%) received argatroban treatment while the

**Funding:** The author(s) received no specific funding for this work.

**Competing interests:** The study was supported by Quzhou City Science and Technology Research Project in 2023 (Grant No. 2023k193), with Dr. Haiyan Xie as the principal investigator.

**Abbreviations:** mRS, modified Rankin Scale; RR, Risk ratio; RCTs, Randomized Controlled Trials; NOS, Newcastle-Ottawa Scale.

remaining 4636(53.7%)did not. The primary outcome of 90-day functional independence (modified Rankin scale (mRS) score≤2) showed that the risk ratio (RR) for patients using argatroban after alteplase thrombolytic therapy compared to those not using argatroban was(RR, 1.00 ([95% CI, 0.92–1.09]; P = 0.97), indicating no statistical significance. However, for patients using argatroban after antiplatelet therapy, was (RR,1.09 [95% CI, 1.04–1.14]; P = 0.0001), which was statistically significant. In terms of hemorrhagic events, the RR for patients using argatroban compared to those not using argatroban was (RR,1.08 [95% CI, 0.88–1.33]; P = 0.46), indicating no statistical significance.

## Conclusion

The results of this study suggest that further research into combination therapy with argatroban and antiplatelet agents may be warranted, however more rigorous RCTs are needed to definitively evaluate the effects of combination treatment.

## Introduction

Globally, a disconcerting phenomenon is unfolding: one in every four adults over the age of 25, one will endure the devastating consequences of stroke in their lifetime, affirming stroke as the second leading cause of death worldwide[1,2]. The severity of this issue accentuates our imperative to develop and refine treatment strategies, that mitigate the burden that stroke imposes on individuals and society[3]. The European Stroke Organization (ESO) has released guidelines for intravenous thrombolysis treatment for acute ischemic stroke, with Alteplase—a thrombolytic agent—proven to be notably effective in the treatment of acute ischemic stroke, deep vein thrombosis, and other vascular diseases [4]. Additionally, antiplatelet drugs such as aspirin, clopidogrel, prasugrel, and ticagrelor, along with phosphodiesterase inhibitors like dipyridamole and cilostazol, have demonstrated significant roles in the prevention and treatment of ischemic stroke and acute coronary syndrome (ACS) [5,6].

Argatroban is a selective anticoagulant that can inhibit thrombin activity. It has been shown to improve neurological symptoms and daily activities in patients with acute ischemic stroke [7,8]. Although argatroban has been proven effective when used alone, it is unclear if additional benefits can be achieved by combining it with antiplatelet or alteplase thrombolytic treatment in acute stroke patients. In order to evaluate the impact of combining argatroban with current antiplatelet or Alteplase treatment strategies, we conducted a systematic review and meta-analysis. The results of this study will provide new insights and data on the potential of this combined treatment strategy.The findings of our study have important clinical implications for the treatment of acute stroke patients, which can lead to a reduction in disease incidence and an improvement in quality of life. Although some positive results have been obtained through the use of argatroban in acute stroke patients, further large-scale randomized controlled trials are required to confirm these findings and provide direction for future research. We will discuss our research methodology, results, and implications in detail in subsequent sections.

Through a meta-analysis and systematic review, our objective is to identify the most effective treatment strategies for acute stroke patients, ultimately improving their quality of life and increasing their survival expectancy. This study is expected to provide strong support for future clinical decisions and drive advancements in acute stroke treatment.

## Methods

### Ethics

This comprehensive meta-analysis was meticulously conducted with strict adherence to the criteria outlined in the Preferred Reporting Items for Systematic Reviews and Meta-analyses [9]. We registered our protocol in PROSPERO with ID: CRD42023428156.

### Search strategy

Two independent scholars embarked on a comprehensive exploration of literature utilizing resources such as PubMed, EMBASE, Web of Science, CENTRAL (The Cochrane Central Register of Controlled Trials), and The Cochrane Library. The time span for this retrieval process extended from the inception of each respective database to April 2023. We included English language foreign literature, without restrictions on the study type or publication year. The exploratory process involved a symbiosis of subject headings and free words. Specific key terms used for sourcing relevant studies from the databases were "argatroban", "Brain Ischemia", "Stroke", "Cerebral Infarction", and "Cerebrovascular Disorders". Both authors independently undertook the tasks of literature search, data collation, and quality appraisal, guided by the inclusion and exclusion criteria, with a subsequent cross-validation of their findings. In addition, to avoid missing any potentially relevant studies the reference lists of the retrieved articles were thoroughly examined.

### Study selection

The criteria for incorporating articles into our study are as follows: (1) Clinical control trials published in English; (2) Patients suffering from acute stroke who have been treated with argatroban, irrespective of their nationality, race, gender, or age. Intervention: The experimental group was treated with argatroban, whereas the control group was not; (4) At least one of the following objective indicators must be included: mRS 0–2 or hemorrhagic events. The exclusion criteria are: (1) Literature published in languages other than English; (2) Redundant literature; (3) Case reports without a control group; fundamental research, such as animal experiments, cadaver studies, biomechanical research, etc.; (4) Literature reviews; (5) Incomplete data from conference literature or literature that cannot be meta-analyzed.

**Outcomes.** The definition of hemorrhagic events as a safety outcome encapsulates any reported bleeding within 90 days in any organ of the body, inclusive of the cranium, skin, mucous membranes, gastrointestinal tract, urine, and gum bleeding, among others. We conducted an assessment of mRS values within three months following argatroban treatment. The mRS scale ranges from 0 (symptom-free) to 6 (death), with an mRS score of 0–2 being defined as an outcome of functional independence [10].

**Data extraction and quality assessment.** During the processes of literature exploration, data procurement, and quality evaluation, any discord between the two authors was addressed through a consensus-based discussion, and if a resolution could not be reached, a third author served as the arbitrator. From the eligible investigations, we extracted pertinent details such as the researchers, publication year, study type, argatroban administration methodology, sample size, gender distribution, mean age, and outcome measures. The Newcastle-Ottawa Scale (NOS) was employed to appraise the quality of cohort investigations. NOS scores, ranging between 0 and 9, were determined based on three aspects: selection, comparability, and outcome. Cohort studies with an NOS score of $\geq 7$ and RCTs were deemed to be of high quality [11].

**Statistical analysis.** We employed Review Manager (Version 5.3 for Windows) and STATA 14.1 for all meta-analyses. The risk ratios (RRs) for dichotomous variables were calculated, with all results reported with a 95% confidence interval (95% CI). The presence of heterogeneity was initially determined using the Q statistic, followed by the estimation of the magnitude of heterogeneity with the $I^2$ value. If P≥0.1 and $I^2$<50%, we considered no significant heterogeneity between groups and integrated effect sizes using a fixed-effects model. Otherwise, a random-effects model was applied. Sensitivity analysis was performed using STATA 14, and we sought the sources of heterogeneity via a literature screening method, culminating in the use of regression analysis. In our meta-analysis, a P-value of <0.05 was deemed statistically significant for differences between groups.

# Results

## Search results and study characteristics

In a comprehensive search across five databases, a total of 1376 relevant articles were retrieved. According to the inclusion and exclusion criteria, a careful review of titles and abstracts was conducted, resulting in 35 articles deemed potentially eligible. Upon further examination of the full texts, 11 articles were ultimately included. The process of literature search and selection is illustrated in Fig 1. The present meta-analysis encompassed three randomized controlled trials and eight retrospective cohort studies. Among these, one study originated from the United States, one from Korea, one from Canada, two from Japan, and six from China. The meta-analysis collectively included 8,635 patients, of which 3,999 belonged to the argatroban group and 4,636 to the control group. The involved literature spanned from 2004 to 2023. The characteristics of the studies and the results of the quality assessment are presented in Tables 1 and 2. All included studies were deemed to be of high quality.

## Safety and efficacy outcomes

**mRS 0–2.** The heterogeneity among the 11 studies included in this research was evaluated using the STATA14 software. The $I^2$ statistic was 59%, and the P-value of the Q-test was 0.007, less than 0.01, indicating statistically significant heterogeneity among the chosen studies. Further inspection of the L'Abbé plot and the Galbraith plot suggested potential strong heterogeneity in a few studies (Fig 2A and 2B), resulting in the determination of moderate heterogeneity among the studies. Consequently, both a random-effects model could be used for synthesis, and the cause of heterogeneity could be explored.

Employing the random-effects model for pooling risk ratios (RR), the final calculated RR was 1.03 (95% Confidence Interval: 0.98–1.09). This result implies that the number of patients with an mRS score of 0–2 after using argatroban was 1.03 times the number of those not using argatroban, a result not statistically significant (Z = 1.22, P = 0.22>0.05). This suggests that there was no significant difference in reducing the mRS 0–2 score in acute stroke patients whether argatroban was used or not (Fig 2D).

A sensitivity analysis conducted on the 11 studies did not reveal any studies that significantly impacted the heterogeneity (Fig 2C).

The studies were divided into three groups based on the control measures: the Alteplase group (Alteplase + argatroban), Antiplatelet group (containing at least one of aspirin and clopidogrel + argatroban), and the Other Medication group (Exclude any treatments for the above two conditions or any results of mRS scores that were not monitored at 90 days). Using the group (control measure) variable as a covariate for meta-regression, the results suggested that the control measure was a source of heterogeneity. Specifically, the Antiplatelet group had

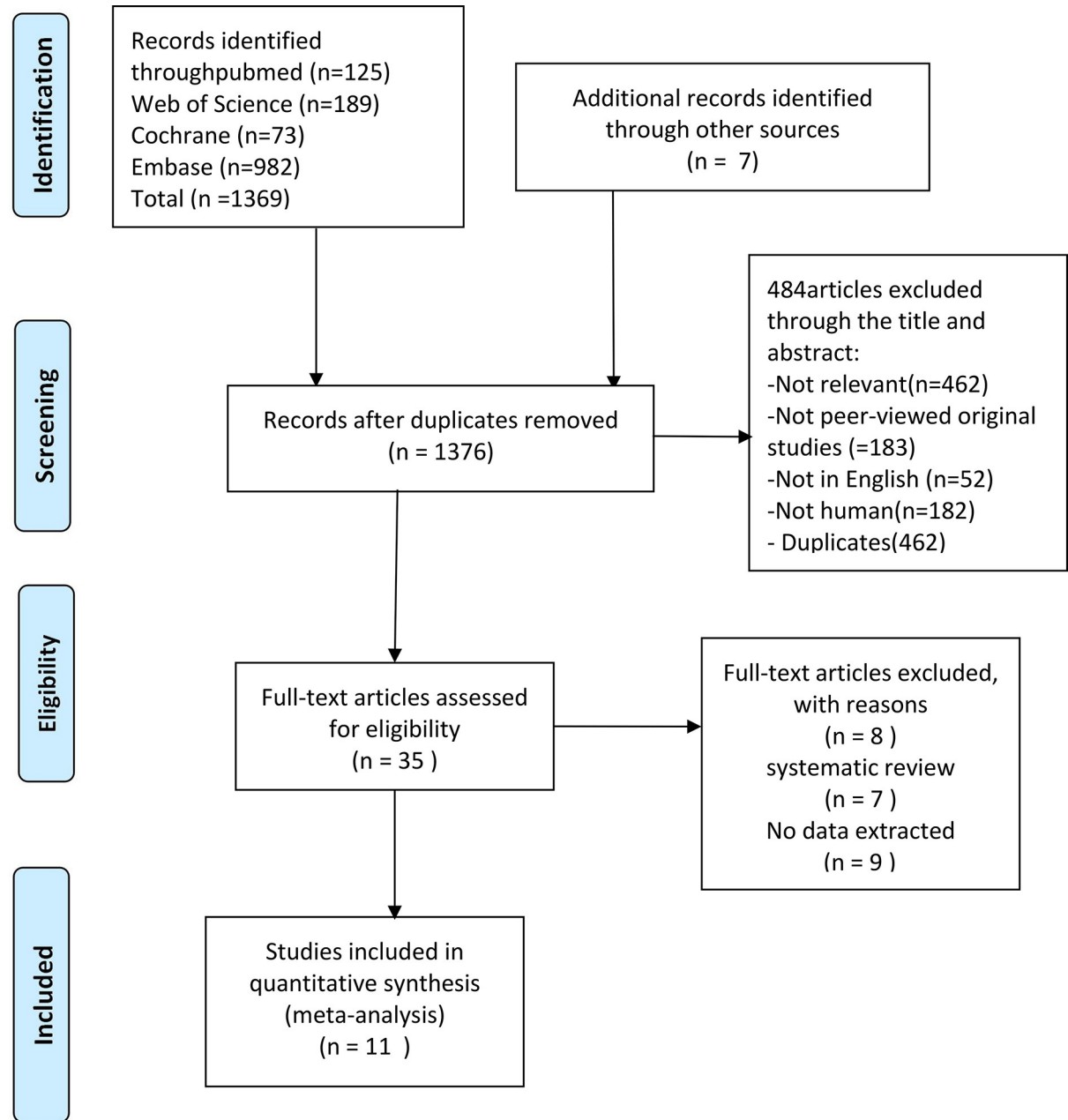

**Fig 1. A flow chart depicting how studies were searched and screened.**

a *P*-value of 0.005, which is less than 0.05, indicating that subgroup analysis should be conducted according to the group (control measure).

Subgroup analysis revealed the following:①For the Alteplase group (Alteplase + argatroban), the heterogeneity ($I^2$ = 0%, P = 0.81>0.05) was not statistically significant. Therefore, a fixed-effect model was chosen to combine the effect sizes, resulting in an RR of 1.00 (0.92–1.09). These results indicate that the use of Alteplase in combination with argatroban in acute stroke patients had no significant effect on the mRS 0–2 score (Z = 0.04, *P* = 0.97).②For the Antiplatelet group (Antiplatele+argatrobant), the heterogeneity ($I^2$ = 0%, P = 0.76>0.1) was not statistically significant. Thus, a fixed-effect model was chosen to combine the effect sizes,

**Table 1. Study characteristics and quality assessment results.**

| Author (year) | Country | Stuty design | Study center | Sample size | Dosage |
|---|---|---|---|---|---|
| Barreto, A D 2017 [12] | USA | RCT | Multiple-center | 59 | Rt-PA bolus followed by argatroban infusion of 1.0μg/kg per minute per minute for 48 hours . |
| Chen,H S 2023 [13] | china | RCT | Multiple-center | 696 | Patients in the argatroban plus alteplase group received a 100-μg/kg intravenous argatroban bolus over 3 to 5 minutes within 1 hour of the alteplase bolus, followed by an argatroban infusion of 1.0 μg/kg per minute for 48 hours. |
| Chen, S 2020 [14] | china | Retrospective cohort | Single-center | 1325 | Oral antiplatelet agents should be administrated as quickly as possible, argatroban 60 mg / day by instillation for 2 days as soon as possible, and argatroban 20 mg / day by instillation for 3 h on 5 consecutive days, 2 times each morning and evening. |
| Kim, J 2019 [15] | Korea | Retrospective cohort | Multiple-center | 133 | patients who were administrated IV t-PA before MT then argatroban began as 100 mg/kg bolus for 3e5 minutes followed by a continuous IV infusion of 3.0 mg/kg/minute for 24 hours |
| LaMonte, M P 2004 [16] | Canada | RCT | Multiple-center | 105 | Argatroban was administered intravenously as an initial 100 g / kg bolus over 3 to 5 minutes as a continuous infusion at a rate of 1 g / kg per minute (low-dose argatroban group), with placebo used in the control group |
| Li, X Q 2022 [17] | China | Retrospective cohort | Single-center | 649 | Within 48 h after onset Antiplatelet therapy was followed by an intravenous injection of 100 μ g / kg argatroban over 3 to 5 minutes, followed by an infusion of 1.0 μ g / kg per minute. |
| Oguro, H 2018 [18] | Japan | Retrospective cohort | Single-center | 513 | Argatroban (120 mg / day) was infused over a 2-hour period, twice daily for 5 days and ozagrel was used in the control group |
| Wada, T 2016 [19] | Japan | Retrospective cohort | Multiple-center | 4578 | argatroban was administered by continuous infusion at a dose of 60 mg / day for the first 2 days and then at a dose of 10 mg twice daily for 3 hours for 5 days, and patients had statin use or alteplase within 7 days. |
| Wang, P F 2021 [20] | China | Retrospective cohort | Multiple-center | 304 | All received antiplatelet therapy, in the argatroban group on admission, with a continuous infusion of 60 mg per day for the first 2 days and then twice daily (20 mg per day) for 5 consecutive days. |
| Xu, J 2022 [21] | China | Retrospective cohort | Single-center | 80 | Antiplatelet and argatroban therapy within 48 hours of admission. |
| Zhou, L S 2020 [22] | China | Retrospective cohort | Single-center | 102 | Argatroban plus antiplatelet therapy with argatroban 60 mg / day as a continuous infusion for 2 days, followed by 20 or 30 mg daily for 2 to 5 days |

resulting in an RR of 1.09 (1.04–1.14). These results indicate that the combination of Antiplatelet therapy and argatroban in acute stroke patients significantly affected the mRS 0–2 score ($Z = 4.02$, $P<0.0001$).③For the Other Medication group (Any other drug or unclear results + argatrobant), the heterogeneity ($I^2 = 37\%$, $P = 0.19>0.1$) was not statistically significant. Therefore, a fixed-effect model was chosen to combine the effect sizes, resulting in an RR of 0.97 (0.93–1.01). These results suggest that the combined use of argatroban with other medications (such as placebos, Ozagrel sodium, etc.) in acute stroke patients had no significant effect on the mRS 0–2 score ($Z = 4.02$, $P = 0.19>0.05$) (Fig 3).

The symmetry of the funnel plots was assessed, yielding the following results: for the Alteplase group ($P = 0.946>0.05$), the Antiplatelet group ($P = 0.054>0.05$), and the Other Medication group ($P = 0.486$). In all three groups, there was no evidence of publication bias.

**Bleeding events.** Heterogeneity testing of the 11 studies included in this research revealed an $I^2$ statistic of 1% and a P-value of 0.95, suggesting no significant heterogeneity among the selected studies. As a result, a fixed-effect model was employed to combine effect sizes.The combined effect size (RR) from the 11 studies using a fixed-effect model was 1.07 (95% Confidence Interval: 0.89–1.29), which was not statistically significant ($Z = 0.74$, $P = 0.46>0.05$). This finding indicates that the use of argatroban in the treatment of acute stroke had no statistically significant effect on hemorrhagic events, as presented in the corresponding figure (Fig 4).

A funnel plot was created to assess potential publication bias among the 11 studies. The symmetry of the funnel plot (Egger's p = 0.216) suggested no publication bias, underscoring the reliability and credibility of the findings of this research.

**Table 2. Bias evaluation.**

| RCTs[♦] | | | | | | | | |
|---|---|---|---|---|---|---|---|---|
| **Study** | Random sequence generation | Allocation concealment | Blinding of participants and personnel | Blinding of outcome assessment | Adequate assessment of incomplete outcome | Selective reporting avoided | No other bias | |
| Barreto, A D 2017 [12] | Low risk | Low risk | Low risk | Low risk | Low risk | Low risk | Low risk | |
| Chen,H S 2023 [13] | Low risk | Low risk | Low risk | Low risk | Low risk | Low risk | Low risk | |
| LaMonte, M P 2004 [16] | Low risk | Low risk | Low risk | Low risk | Low risk | Low risk | Low risk | |
| Non-RCTs[★] | | | | | | | | |
| Study | Selection(score) Representativeness of the exposed cohort | Selection of the non-exposed cohort | Ascertainment of exposure | Demonstration that outcome of interest was not present at start of study | Comparability Comparability of cohorts on the basis of the design or analysis | Exposure (score) Assessment of outcome | Follow-up long enough for outcomes to occur | Adequacy of follow up of cohorts |
| Chen, S 2020 [14] | 1 | 1 | 1 | 1 | 1 | 1 | 0 | 1 |
| Kim, J 2019 [15] | 1 | 1 | 1 | 1 | 1 | 1 | 0 | 1 |
| Li, X Q 2022 [17] | 1 | 0 | 1 | 1 | 1 | 1 | 0 | 1 |
| Oguro, H 2018 [18] | 1 | 1 | 0 | 1 | 1 | 1 | 0 | 1 |
| Wada, T 2016 [19] | 1 | 1 | 1 | 1 | 1 | 1 | 1 | 1 |
| Wang, P F 2021 [20] | 1 | 1 | 1 | 1 | 1 | 1 | 1 | 1 |
| Xu, J 2022 [21] | 1 | 1 | 1 | 1 | 1 | 1 | 0 | 1 |
| Zhou, L S 2020 [22] | 1 | 1 | 1 | 1 | 1 | 1 | 0 | 1 |

## Discussion

Argatroban, alteplase, and antiplatelet agents each possess distinct pharmacological properties that contribute to their effectiveness in treating acute stroke [23–25]. Argatroban, a direct thrombin inhibitor, is advantageous in the prevention of thrombosis in heparin-induced thrombocytopenia due to its short half-life and predictable anticoagulant response [26]. Alteplase, the only thrombolytic agent approved for the treatment of acute ischemic stroke, has been clinically in use since 1986 [27]. Antiplatelet drugs, on the other hand, serve as the foundation for managing cardiovascular disease by preventing thrombosis through inhibiting platelet aggregation [28]. However, while the individual effects of these agents are well-established, further research is needed to assess the efficacy of combining argatroban with alteplase or antiplatelet agents for the treatment of acute stroke [29].

S1 Fig provides additional insight into the heterogeneity of treatment effects of combination therapy with argatroban and different treatment modalities. In our meta-regression analysis, we observed a statistically significant difference between antiplatelet groups ($P = 0.005 < 0.05$), suggesting that there may be variations in the effectiveness of combining argatroban with different treatments. Notably, these supplementary figures provide additional context and support for the main findings of our study. They highlight the heterogeneity of treatment effects and underscore the necessity for additional research to optimize the combination of argatroban with different treatment modalities.

In our latest study, we observed that patients with acute stroke treated with alteplase did not experience a significant functional improvement at 90 days, even if they continued taking argatroban at follow-up. Alteplase, a recombinant tissue plasminogen activator, catalyzes the formation of plasmin at the thrombus site to dissolve the thrombus and restore blood supply [30]. However, the effect of alteplase wanes once blood flow is restored, and the anticoagulant

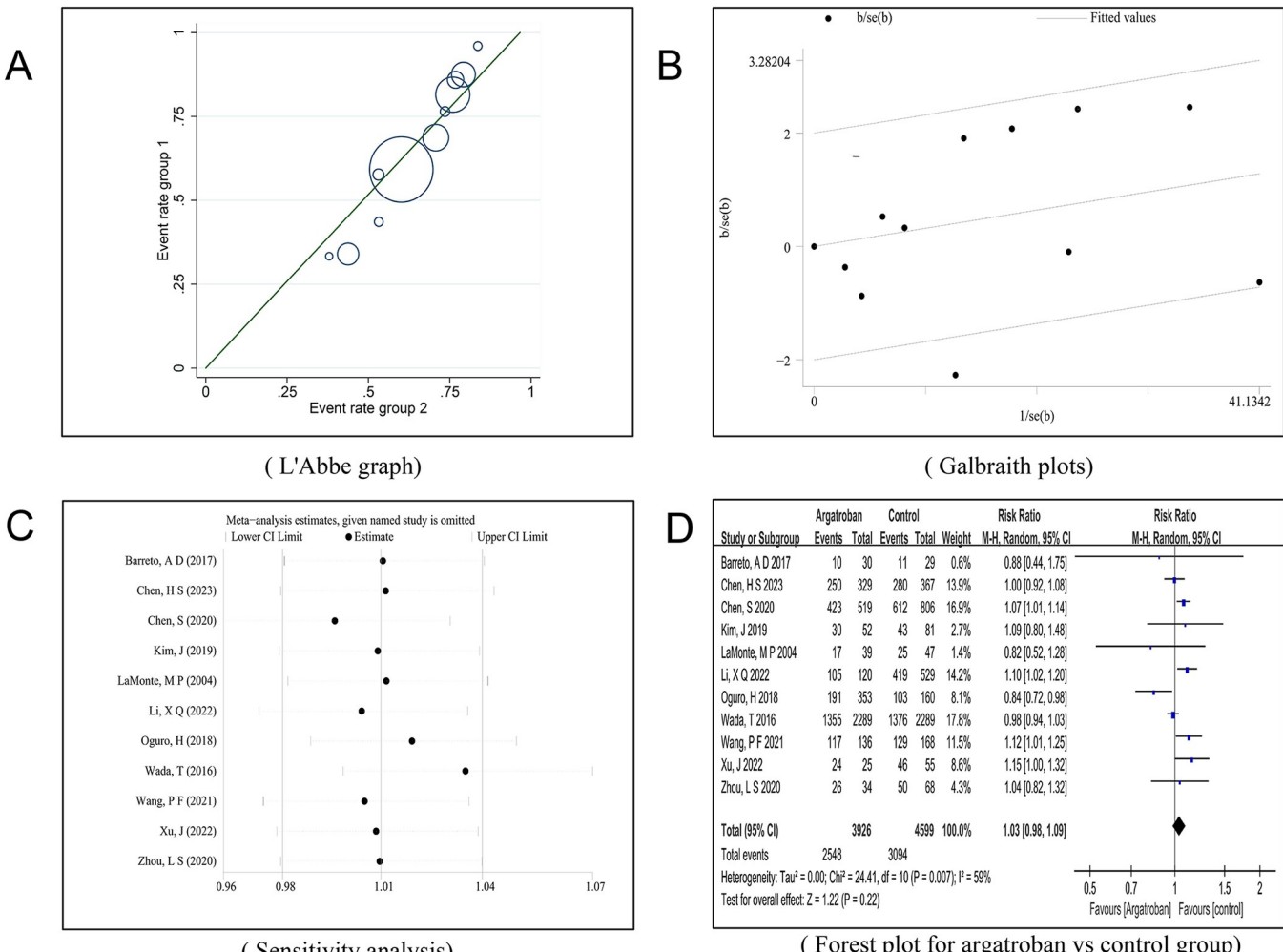

**Fig 2. Results and forest plot for exploring heterogeneity in argatroban vs control group.**

effect of argatroban may not produce a significant ameliorative effect on the already restored blood flow [31]. Consequently, the combination of alteplase and argatroban may not significantly improve functional recovery in patients with acute stroke. On the other hand, antiplatelet drugs prevent platelet aggregation and new thrombus formation, which may be more critical in the early stages of thrombus formation [32]. The anticoagulant effect of argatroban may have a synergistic effect with antiplatelet drugs, improving functional recovery to some extent. However, our study found no significant effect of argatroban on 90-day functional recovery in acute stroke patients compared with ozagrel sodium or placebo. We also considered functional recovery within 7 days of antiplatelet drug use and placed this study in the "other methods" group for more reliable results [22]. Although we attempted to include this result in the antiplatelet group for comparison, we saw no significant changes. Study heterogeneity was paradoxically larger in other methodological groups, and their results require further validation due to the diversity of comparison subjects. Among the available studies, one study has noted that the infusion of argatroban could be an effective and safe treatment option for improving functional outcomes in stroke patients. The study did not indicate whether there was an increased benefit from a combination of different drugs. Nonetheless, the bleeding

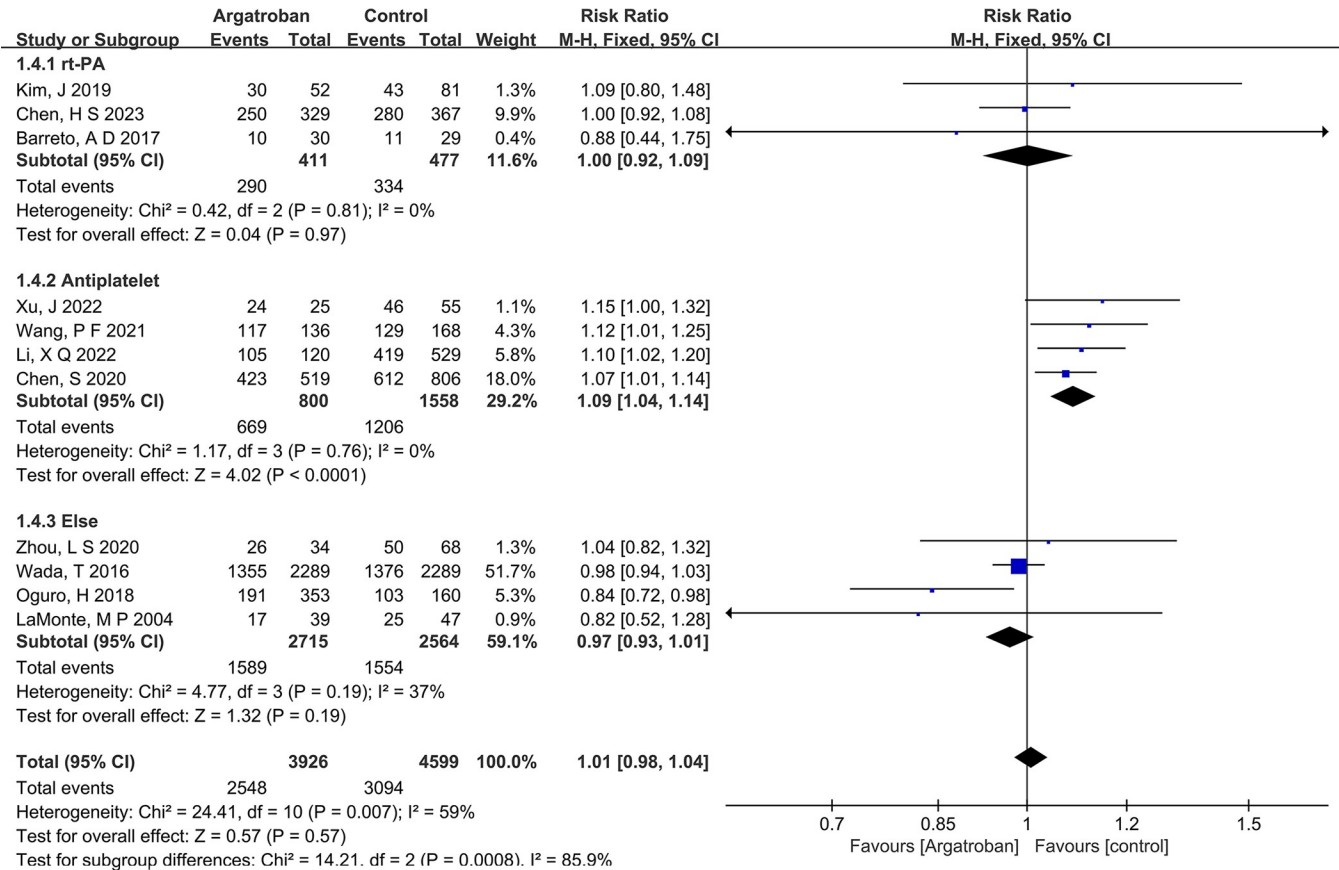

**Fig 3. Forest plot of mRS 0–2 subgroup analysis.**

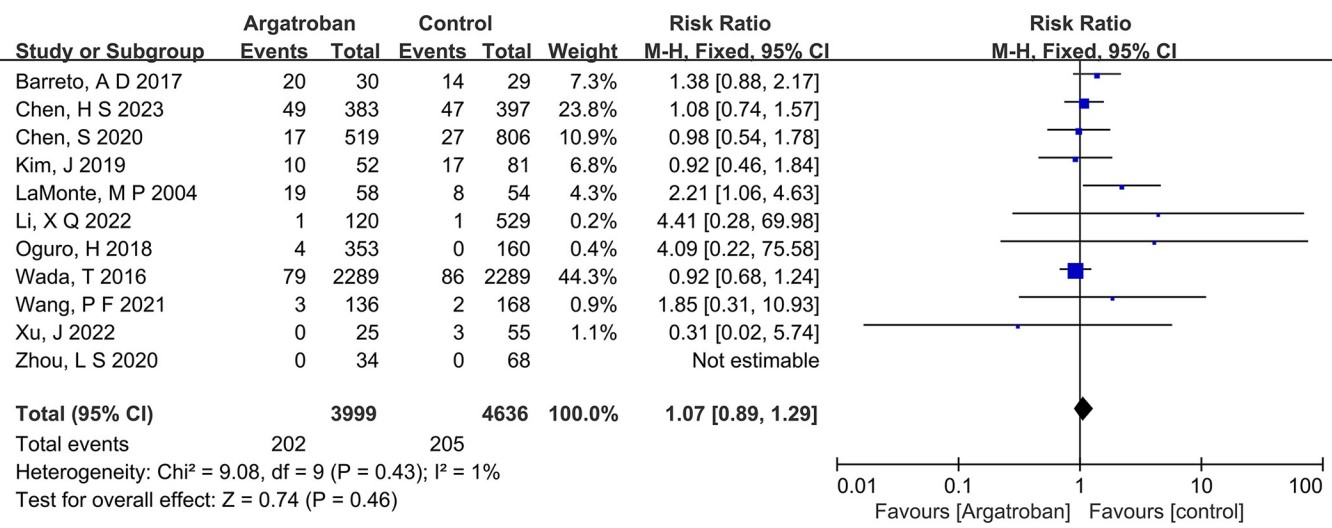

**Fig 4. Forest plot of bleeding events.**

event outcomes remained consistent. Regarding bleeding risk, both alteplase and argatroban can increase the risk, but our study did not observe a significant increase in bleeding risk [33,34]. This may be because physicians adequately considered the risk of bleeding during patient screening and management and used these drugs only in patients with a low bleeding risk. Alternatively, physicians promptly recognized and managed bleeding symptoms during medication. We observed a significant difference in bleeding between the alteplase alone group and the argatroban group. The investigator determined that argatroban did not contribute to the increased risk of bleeding after thrombolysis with alteplase. In our safety study, we considered all possible risks of bleeding in actual clinical work because physicians need to stop all medications, including antiplatelet agents and argatroban, if a patient experiences possible bleeding [35].

There are several important limitations that must be considered when interpreting our findings. Firstly, only three of the included studies were randomized controlled trials, which is the study design that provides the strongest evidence. However, the remaining studies were retrospective non-randomized controlled studies, which may have had selection bias and confounding factor bias, potentially impacting our results. Secondly, as a systematic review, our study relied on already published studies which may have led to publication bias, as studies with negative results are typically more difficult to publish. Furthermore, we did not have access to unpublished data that may have impacted our results. In addition, our study showed some heterogeneity, but we were able to identify the source of heterogeneity through a regression model. However, due to limitations in the number of articles and their content, it was not feasible to conduct further research on potential influencing factors such as baseline data, variations in disease severity, dosages and timing of treatments, and additional therapeutic approaches.Finally, our findings are based primarily on observational studies and cannot determine causality. Therefore, while our study provides valuable insights, these limitations must be taken into account when interpreting our results. In the future, more high-quality randomized controlled trials and observational studies will be necessary to further confirm our findings.

## Conclusions

This study presents a novel treatment strategy for acute stroke patients. Administering argatroban after antiplatelet therapy resulted in a significant improvement in mRS scoresof 0–2, indicating the efficacy of this approach in enhancing functional recovery. In contrast, the use of argatroban after thrombolysis with alteplase did not result in a significant change in mRS scores of 0–2. These findings provide valuable insights for clinicians when choosing treatment strategies and highlight the importance of considering patient-specific circumstances, including whether they have received thrombolytic treatment with alteplase.

## Supporting information

**S1 Fig. Meta-regression results.**
(PNG)

**S2 Fig. Funnel plot of mRS scores 0–2 in subgroups of alteplase or antiplatelet agents plus argatroban.**
(TIF)

**S3 Fig. Funnel plot of Bleeding events in subgroups of alteplase or antiplatelet agents plus argatroban.**
(TIF)

**S1 Table. Search engine and query.**
(DOC)

**S1 File. PRISMA checklist.**
(DOCX)

## Acknowledgments

The authors thank Jinlong Xu, Caimu Wang, Chen Bao, who have been a source of encouragement and inspiration.

## Author Contributions

**Conceptualization:** Haiyan Xie, Zhiyong Lan.

**Data curation:** Wukun Ge, Xiuping Xu.

**Funding acquisition:** Haiyan Xie, Ying Chen, Zhiyong Lan.

**Investigation:** Xiuping Xu, Yina Yang.

**Resources:** Haiyan Xie.

**Supervision:** Wukun Ge, Yina Yang.

**Validation:** Chengjiang Liu.

**Visualization:** Chengjiang Liu.

**Writing – original draft:** Wukun Ge, Chengjiang Liu.

**Writing – review & editing:** Yina Yang.

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
