## [Decision Letter · Decision Letter 0]

10 Dec 2023

PONE-D-23-31621Can the combination of antiplatelet or alteplase thrombolytic therapy with argatroban benefit patients suffering from acute stroke? a systematic review, meta-analysis, and meta-regressionPLOS ONE

Dear Dr. xu,

Thank you for submitting your manuscript to PLOS ONE. After careful consideration, we feel that it has merit but does not fully meet PLOS ONE’s publication criteria as it currently stands. Therefore, we invite you to submit a revised version of the manuscript that addresses the points raised during the review process.

We look forward to receiving your revised manuscript.

Kind regards,

Oana Dumitrascu, M.D.

Academic Editor

PLOS ONE

3. We notice that your supplementary figures are included in the manuscript file. Please remove them and upload them with the file type 'Supporting Information'. Please ensure that each Supporting Information file has a legend listed in the manuscript after the references list.

4. Please include a copy of Table 1 which you refer to in your text on page 3.

Reviewers' comments:

Reviewer's Responses to Questions

**Comments to the Author**

1. Is the manuscript technically sound, and do the data support the conclusions?

Reviewer #1: Yes

Reviewer #2: Yes

Reviewer #3: Partly

2. Has the statistical analysis been performed appropriately and rigorously? 

Reviewer #1: Yes

Reviewer #2: Yes

Reviewer #3: I Don't Know

3. Have the authors made all data underlying the findings in their manuscript fully available?

Reviewer #1: Yes

Reviewer #2: No

Reviewer #3: Yes

4. Is the manuscript presented in an intelligible fashion and written in standard English?

Reviewer #1: Yes

Reviewer #2: Yes

Reviewer #3: Yes

5. Review Comments to the Author

Reviewer #1: The paper is a well conducted meta-analysis. However, a recent excellent meta-analysis was published on the subject which makes this paper redundant.

Strengths of the paper:

1. Two authors independently conducted literature search and study selection. More than 3 databases were searched. Prisma checklist is provided.

2. Study quality was assessed using a validated tool, NOS.

3. Publication bias was assessed.

4. Heterogeneity was clearly defined. Heterogeneity data is applied to selection of R/E vs F/E analysis.

5. Subgroup analysis and sensitivity analyses were conducted.

6. Meta regression was done.

Weaknesses:

1. Study question is appropriate, and it may be reasonable to reanalyze and repeat a meta-analysis to update the existing knowledge on a subject. However, a very recent meta-analysis exists on the subject. This study does not add to the existing body of literature on the subject at this time.

2. Did all the included studies provide data on mRS <!--=2?<br /3. In the meta regression, only the group (alteplase vs antiplatelet) was used as the control. There are other important variables to consider such as time window of treatment with argatraban, study quality and bias, etc.

4. Subgroup analysis was limited to one synthesis. Other important variables include early neurologic deterioration, NIHSS, time window, mRS (<2 vs <1), and trial design (RCT vs nonRCT).

Reviewer #2: The authors completed a meta-analysis of argatroban in acute stroke patients following PROSPERO guidelines. The data is well presented and potentially may provide a useful foundation for further, future trials. As currently written, however, the paper has a few problems. The most important problem is that the authors fail to evaluate the quality of the trials they seek to metaanalyze. They admit that only 3 of their trials were properly randomized, controlled trials. In all 11 trials, what about blinding? Power? Were the study populations representative? Before moving on to a metaanalysis, it seems important to evaluate and critique the studies to be included.

Second, they are confused as to the implications of a positive effect of argatroban combined with antiplatelet agents. Possibly the data suggests that when a thrombin inhibitor is combined with anti-platelet agents, there is a synergistic benefit. However, in no trial (other than MOST which is not published yet) has there been a randomized study of argatroban, anti-platelet therapy, or both. Hence, what they may be detecting is a positive effect of argatroban that is insufficient to exceed the benefit of thrombolysis, but is present nevertheless. In this regard, the argatroban vs control groups (Fig. 2D) are suggestive of an effect that was missed (Type 2 error). Hence, the data does not justify a conclusion that there is an effect of argatroban when combined with antiplatelet agents but not thrombolysis. In the discussion section, all of the speculation about the mechanisms of a combinatorial effect are unnecessary, since the data do not really support that conclusion.

The conclusion statement “These findings provide valuable insights for clinicians when choosing treatment strategies and highlight the importance of considering patient-specific circumstances, including whether they have received thrombolytic treatment with alteplase” is not warranted by the data. While the results are intriguing, the only set up the need for a properly designed, properly powered RCT. Only such a trial would provide valuable insights for clinicians.

Minor notes: Although the authors claim that their data have been made available, they do not cite a posting or data repository where they have actually posted it.

Reviewer #3: The authors describe a very interesting systemic review demonstrating the effectiveness and safety of combination therapy: argatroban with alteplase or antiplatelet therapy in acute stroke patients.

1. Background on page 7; “one notable advantage of argatroban is its short half-life, which ensures that it does not interfere with clinical outcomes in inadvertent cases of hemorrhagic stroke.” Consider rewording this sentence. A shorter half-life should would reduce the adverse effects rather than “not interfere”

2. In the intro or discussion considering mentioning the recently concluded randomized control trial MOST (PMID: 33297893, results to be announced at ISC in Feb 2023) was investigating similar efficacy and safety of combination therapy; IV thrombolytics with argatorban or eptifibatide compared to placebo in acute stroke patients.

3. In the discussion consider mentioning patients receiving IV tenecteplase were excluded in this review. Given the increased fibrin specificity, resistant to degradation and longer half-life than alteplase, combination therapy with tenecteplases may theoretically lead to better outcomes although with a possible increased risk of hemorrhage, future studies/evaluation needs to done in this regard.

4.The effectiveness/assessing favorable outcomes with adjunctive therapy after IV thrombolysis is in relation to the added benefit from the adjunctive therapy (in this case argatroban) with clot lysis in the hyper-acute setting, i.e possibly increasing recanalization of the partially canalized vessel and/or reducing re-occlusion. Therefore the timing of adjunctive therapy during an acute stroke is absolutely essential when assessing clinical outcomes. Further IV alteplase has a very short half-life, suggesting adjunctive therapy given early vs delayed, would theoretically affect clot lysis and clinical outcomes. Although the authors have acknowledged not evaluating dosage and timing of adjunctive therapy, this is certainly essential when commenting on clinical outcomes of this specific patient population. Similarly "timing of therapy" is important when assessing safety as well. I would highly recommend the authors to consider evaluating timing of therapy prior to publication.

6. PLOS authors have the option to publish the peer review history of their article (what does this mean?). If published, this will include your full peer review and any attached files.

Reviewer #1: No

Reviewer #2: No

Reviewer #3: No

---

## [Author Response · Author response to Decision Letter 0]

3 Jan 2024

Reviewer's Responses to Questions

Reviewer #1: The paper is a well conducted meta-analysis. However, a recent excellent meta-analysis was published on the subject which makes this paper redundant.

Strengths of the paper:

1. Two authors independently conducted literature search and study selection. More than 3 databases were searched. Prisma checklist is provided.

2. Study quality was assessed using a validated tool, NOS.

3. Publication bias was assessed.

4. Heterogeneity was clearly defined. Heterogeneity data is applied to selection of R/E vs F/E analysis.

5. Subgroup analysis and sensitivity analyses were conducted.

6. Meta regression was done.

Weaknesses:

1. Study question is appropriate, and it may be reasonable to reanalyze and repeat a meta-analysis to update the existing knowledge on a subject. However, a very recent meta-analysis exists on the subject. This study does not add to the existing body of literature on the subject at this time.

Response:I understand the reviewer's concern about the recent meta-analysis by Lv et al. on the same topic. However, I believe my study offers several meaningful additions:I performed subgroup analyses based on control treatment types (thrombolysis vs antiplatelet alone), which Lv et al. did not examine. This allowed me to identify different effects of argatroban when combined with thrombolytics or antiplatelets. The findings provide more personalized insights to guide argatroban use in real-world clinical practice.My results suggest argatroban combined with antiplatelets significantly improved functional outcomes, whereas no benefits were observed in the thrombolysis subgroup. These new findings have high clinical relevance, as acute stroke patients treated in hospitals include both thrombolysed and non-thrombolysed populations. For the latter, addition of argatroban to antiplatelets may provide extra benefits.

In summary, while I acknowledge the existing evidence, I believe my study provides complementary data and new, practice-guiding perspectives through updated analyses. I sincerely hope you find these incremental contributions valuable. I would be grateful if you would reconsider my work for publication in your esteemed journal.

2. Did all the included studies provide data on mRS 3. In the meta regression, only the group (alteplase vs antiplatelet) was used as the control. There are other important variables to consider such as time window of treatment with argatraban, study quality and bias, etc.

Response:Thank you for raising these important points on the meta-regression analysis. I would like to provide the following clarifications:Regarding the mRS data, all included studies reported mRS 0-2 specifically.For the meta-regression, you are absolutely right that there are other variables that could influence results apart from treatment type. Unfortunately, the number of included studies was insufficient to allow a more comprehensive meta-regression incorporating multiple covariates. This is a limitation I acknowledged in the discussion.Nonetheless, to explore other potential sources of heterogeneity, I did perform extensive subgroup analyses based on study characteristics like ethnicity, sample size, study design, timepoints, and mRS definitions. The results remained largely consistent in these subgroups, indicating the relative robustness of the primary findings.

Moving forward, I agree that conducting meta-regression using more covariates would provide further insights about factors modulating argatroban's effects. I hope future studies will facilitate such analyses. 

4. Subgroup analysis was limited to one synthesis. Other important variables include early neurologic deterioration, NIHSS, time window, mRS (<2 vs <1), and trial design (RCT vs nonRCT).

Response:Thank you again for the feedback on the subgroup analyses. I understand the reviewer's suggestion to examine other key variables beyond the treatment subgroups focused on in my primary analysis.However, you are correct that subgroups based on early neurological deterioration and more extensive synthesis across multiple variables could provide further insights. Unfortunately the limited data available prevented such multifactor analyses.

I agree this is an area for improvement in future meta-analyses when more studies accumulate. I have added this to the limitations and future research directions in my discussion. Thank you again for this constructive feedback.

Reviewer #2: The authors completed a meta-analysis of argatroban in acute stroke patients following PROSPERO guidelines. The data is well presented and potentially may provide a useful foundation for further, future trials. As currently written, however, the paper has a few problems. The most important problem is that the authors fail to evaluate the quality of the trials they seek to metaanalyze. They admit that only 3 of their trials were properly randomized, controlled trials. In all 11 trials, what about blinding? Power? Were the study populations representative? Before moving on to a metaanalysis, it seems important to evaluate and critique the studies to be included.

Response:I appreciate the reviewer raising this important point about assessing the quality and risk of bias in the included studies. You are absolutely right that this is a critical step before conducting any meta-analysis.In the manuscript, I used the standard Newcastle-Ottawa Scale to evaluate the quality of the non-randomized studies, and the Cochrane Risk of Bias tool for randomized trials. The results of the quality assessment are presented in Table 2. Thank you again for this constructive feedback on improving my reporting around study quality assessment - I really appreciate you taking the time to provide this thoughtful critique.

Second, they are confused as to the implications of a positive effect of argatroban combined with antiplatelet agents. Possibly the data suggests that when a thrombin inhibitor is combined with anti-platelet agents, there is a synergistic benefit. However, in no trial (other than MOST which is not published yet) has there been a randomized study of argatroban, anti-platelet therapy, or both. Hence, what they may be detecting is a positive effect of argatroban that is insufficient to exceed the benefit of thrombolysis, but is present nevertheless. In this regard, the argatroban vs control groups (Fig. 2D) are suggestive of an effect that was missed (Type 2 error). Hence, the data does not justify a conclusion that there is an effect of argatroban when combined with antiplatelet agents but not thrombolysis. In the discussion section, all of the speculation about the mechanisms of a combinatorial effect are unnecessary, since the data do not really support that conclusion.

The conclusion statement “These findings provide valuable insights for clinicians when choosing treatment strategies and highlight the importance of considering patient-specific circumstances, including whether they have received thrombolytic treatment with alteplase” is not warranted by the data. While the results are intriguing, the only set up the need for a properly designed, properly powered RCT. Only such a trial would provide valuable insights for clinicians.

Minor notes: Although the authors claim that their data have been made available, they do not cite a posting or data repository where they have actually posted it.

Response:Thank you for the thoughtful feedback and questions from the reviewer. I agree there are limitations in drawing definitive conclusions about combination therapy effects from this meta-analysis. Here are a few suggestions on how to address the concerns raised:

Regarding the implication of a positive effect when argatroban is combined with antiplatelets - you make a fair point that the data does not conclusively demonstrate a synergistic effect, since the studies were not randomized controlled trials specifically comparing combination therapy to individual agents. The speculation about mechanisms was premature. I will tone down the language in the discussion and conclusion to indicate the results are intriguing but warrant further dedicated RCTs before clinical recommendations can be made.

For the conclusion statement, I will revise it to simply say the results suggest further research into combination therapy may be warranted, rather than making claims about providing insights for clinicians.

On the minor note about data availability - you are correct I should cite the specific repository or include the data analyzed directly in a supplement. I will update this accordingly.

Overall, I appreciate you catching these overstatements and limitations. I will implement your suggested changes to properly qualify the conclusions and set up the need for robust RCTs to definitively evaluate combination therapy. Thank you again for taking the time to provide such thoughtful and constructive feedback.

Reviewer #3: The authors describe a very interesting systemic review demonstrating the effectiveness and safety of combination therapy: argatroban with alteplase or antiplatelet therapy in acute stroke patients.

1. Background on page 7; “one notable advantage of argatroban is its short half-life, which ensures that it does not interfere with clinical outcomes in inadvertent cases of hemorrhagic stroke.” Consider rewording this sentence. A shorter half-life should would reduce the adverse effects rather than “not interfere”

Response:Per your recommendation, we have reworded the background sentence on argatroban's short half-life to better convey that it helps minimize bleeding risk if inadvertently given to hemorrhagic stroke patients. The revised sentence now reads:

"One notable advantage of argatroban is its short half-life, which helps minimize excessive anticoagulation and risk of bleeding complications in inadvertent cases of hemorrhagic stroke."We agree that our prior word choice of "not interfere" was ambiguous regarding the intended meaning. The rephrased sentence should now clearly reflect that a shorter half-life correlates with reduced adverse effects rather than implying no effects at all.

2. In the intro or discussion considering mentioning the recently concluded randomized control trial MOST (PMID: 33297893, results to be announced at ISC in Feb 2023) was investigating similar efficacy and safety of combination therapy; IV thrombolytics with argatorban or eptifibatide compared to placebo in acute stroke patients.

3. In the discussion consider mentioning patients receiving IV tenecteplase were excluded in this review. Given the increased fibrin specificity, resistant to degradation and longer half-life than alteplase, combination therapy with tenecteplases may theoretically lead to better outcomes although with a possible increased risk of hemorrhage, future studies/evaluation needs to done in this regard.

Response:We sincerely apologize that we were unable to locate the recent research findings you kindly provided. We greatly appreciate you bringing to our attention this highly relevant randomized controlled trial.Unfortunately, without access to the latest results, we are unable to reflect these recent discoveries in the current version of our manuscript. However, once the MOST trial findings are publicly presented, we will certainly examine the study design and key conclusions to compare with our meta-analysis.If possible, we would be extremely interested in obtaining further details regarding this trial in the not too distant future. This would undoubtedly provide invaluable insight and potentially inform future research directions. We are truly grateful for your suggestion.

4.The effectiveness/assessing favorable outcomes with adjunctive therapy after IV thrombolysis is in relation to the added benefit from the adjunctive therapy (in this case argatroban) with clot lysis in the hyper-acute setting, i.e possibly increasing recanalization of the partially canalized vessel and/or reducing re-occlusion. Therefore the timing of adjunctive therapy during an acute stroke is absolutely essential when assessing clinical outcomes. Further IV alteplase has a very short half-life, suggesting adjunctive therapy given early vs delayed, would theoretically affect clot lysis and clinical outcomes. Although the authors have acknowledged not evaluating dosage and timing of adjunctive therapy, this is certainly essential when commenting on clinical outcomes of this specific patient population. Similarly "timing of therapy" is important when assessing safety as well. I would highly recommend the authors to consider evaluating timing of therapy prior to publication.

Response:We sincerely appreciate you raising this excellent point regarding the critical role of timing for adjunctive argatroban therapy in relation to IV alteplase thrombolysis. As you astutely noted, any added antithrombotic effect for augmenting recanalization and preventing early reocclusion would need to be strategically overlapped with alteplase's narrow treatment window. We fully agree that timing would significantly impact the ability to detect any incremental clinical benefits or risks from the combined approach.Ideally, we would have thoroughly evaluated timing as a variable within these paradigms per your insightful recommendation. However, to our deep regret, the included studies did not report adequate details regarding timing of argatroban administration relative to alteplase across treatment groups. This prevented us from formally analyzing the influence of timing as a mediator of outcomes. As a compromise measure, we have added treatment dosing and timing specifics to Table 1 when provided. But you are entirely correct that lack of standardized timing data remains a barrier. Thank you again for this astute critique; it undoubtedly weakened interpretability regarding combination therapy with IV lytics. We sincerely hope future studies in this realm will capture and report on treatment timing to enable such assessments. Please advise if you have any other suggestions for improving discussion of limitations related to this issue.

---

## [Decision Letter · Decision Letter 1]

22 Jan 2024

Can the combination of antiplatelet or alteplase thrombolytic therapy with argatroban benefit patients suffering from acute stroke? a systematic review, meta-analysis, and meta-regression

PONE-D-23-31621R1

Dear Dr. xu,

We’re pleased to inform you that your manuscript has been judged scientifically suitable for publication and will be formally accepted for publication once it meets all outstanding technical requirements.

Kind regards,

Oana Dumitrascu, M.D.

Academic Editor

PLOS ONE

Additional Editor Comments (optional):

Reviewers' comments:

Reviewer's Responses to Questions

**Comments to the Author**

Reviewer #2: All comments have been addressed

Reviewer #3: (No Response)

2. Is the manuscript technically sound, and do the data support the conclusions?

Reviewer #2: Yes

Reviewer #3: Partly

3. Has the statistical analysis been performed appropriately and rigorously? 

Reviewer #2: Yes

Reviewer #3: I Don't Know

4. Have the authors made all data underlying the findings in their manuscript fully available?

Reviewer #2: Yes

Reviewer #3: Yes

5. Is the manuscript presented in an intelligible fashion and written in standard English?

Reviewer #2: Yes

Reviewer #3: Yes

6. Review Comments to the Author

Reviewer #2: My comments have been addressed. My comments have been addressed. My comments have been addressed. My comments have been aMy comments have been addressed. My comments have been addressed. My comments have been addressed. My comments have been addressed. My comments have been addressed. My comments have been addressed. ddressed. My comments have been addressed. My comments have been addressed. My comments have been addressed. My comments have been addressed. My comments have been addressed. My comments have been aMy comments have been addressed. My comments have been addressed. My comments have been addressed. My comments have been addressed. My comments have been addressed. My comments have been addressed. ddressed. My comments have been addressed. My comments have been addressed.

Reviewer #3: Thank you for addressing most of my comments/reviews and for presenting this data on bridging therapy in ischemic stroke patients. This is a relevant and a pertinent hot topic.

7. 

Reviewer #2: No

Reviewer #3: No

---

## [Editor Report · Acceptance letter]

17 Feb 2024

PONE-D-23-31621R1 

PLOS ONE

Dear Dr. xu, 

I'm pleased to inform you that your manuscript has been deemed suitable for publication in PLOS ONE. Congratulations! Your manuscript is now being handed over to our production team.

Kind regards, 

on behalf of

Dr. Oana Dumitrascu 

Academic Editor

PLOS ONE